# Study of the Feasibility of Proposed Measures to Assess Animal Welfare for Zebu Beef Farms within Pasture-Based Systems under Tropical Conditions

**DOI:** 10.3390/ani13233659

**Published:** 2023-11-27

**Authors:** Marlyn H. Romero, Jhoan Barrero-Melendro, Jorge A. Sanchez

**Affiliations:** 1Department of Animal Health, Faculty of Agrarian and Animal Sciences, Universidad de Caldas, Manizales 170004, Colombia; jorge.sanchez@ucaldas.edu.co; 2CIENVET Group, Faculty of Agrarian and Animal Sciences, Universidad de Caldas, Manizales 170004, Colombia; jhoan.barrero28199@ucaldas.edu.co

**Keywords:** animal wellbeing, beef cattle, good health, welfare assessment

## Abstract

**Simple Summary:**

Consumers perceive pasture-based livestock production as natural and ethical and, therefore, better for animal welfare. The objective of the study was to test the feasibility of the proposed measures to assess animal welfare on 24 commercial zebu cattle farms in tropical pasture systems. The methodology was developed through participatory workshops with producers, academia, and health authorities. The methodology included animal-based, resource-based, and management-based indicators. Study of the on-farm feasibility of the measures was carried out through interviews, the review of records, the direct observation of animal lots, and individual evaluation in the pastures. Application of the methodology and expert analysis demonstrated that simple measures exist to assess animal welfare in pasture systems. The protocol will help identify opportunities for improvement to strengthen the implementation of animal welfare practices and comply with sanitary requirements.

**Abstract:**

Pasture-based production systems are predominant in major beef-producing countries; however, these systems lack validated protocols to assess animal welfare under commercial conditions. The objective of this study was to test the feasibility of the proposed measures and methodology for the evaluation of animal welfare in fattening cattle under pasture conditions. The initial methodology was developed with the participation of producers, professionals, the general public, and the Colombian health authority, through workshops with a participatory approach and collaborative knowledge management. The study was carried out in 24 pasture-based commercial Zebu cattle farms in the middle Magdalena region of Colombia. Visits were made with an average duration of 2.5 h, which included the evaluation of 788 fattening cattle. The methodology evaluated animal-based, resource-based, and management-based indicators through a questionnaire-guided interview to evaluate cattle handling and health, animal-based measurements, and documentation management. A protocol validation process was carried out by selecting indicators that remained unchanged, adjusting those that were feasible to implement, and removing inadequate indicators. The application of the methodology demonstrated that there are feasible measures to include in the evaluation protocols of pasture-based fattening systems. Likewise, the active participation of producers is crucial to achieving a greater commitment to the implementation of this methodology for the assessment of animal welfare in cattle under pasture conditions.

## 1. Introduction

The intensification of animal production has increased public awareness of environmental conservation, health, and welfare, aimed at promoting food safety, food security, and sustainable food production [1,2]. Consumers perceive pasture-based livestock production as natural and ethical, thereby better for animal welfare, compared to confined systems [3,4]. Likewise, consumers are willing to pay more for milk and meat from pasture-raised cattle [5]. However, there does seem to be a difference between what consumers say in surveys and what their actual buying habits are. In surveys, they say they are willing to pay more for animal welfare standard products, but in practice, they sometimes buy the cheaper option. This habit and the difference between what consumers say in surveys and what happens must be studied and considered [6].

The increasing focus on ensuring that animals have “lives worth living” makes pasture-based production systems important for cattle to develop positive emotions, favoring their natural behavior [7,8]. Similarly, cattle have a greater preference for natural pasture-based environments [9], in which they develop a more efficient immune system [10], can exercise more, and maintain social cohesion; they present less risk of hoof injuries, lameness, culling and mortality, but a higher risk of internal parasitism, biosecurity problems (greater contact with wild animals) [7,11] and thermal comfort [3,4], among others.

Protocols have been proposed to assess the welfare of cattle in confinement (Welfare Quality^®^, AssureWel, and others), but few are aimed at evaluating extensive or pasture-based Zebu beef cattle farms [12], which address all the welfare challenges that animals must face in these systems. Worldwide, studies show that a large proportion of dairy cows are raised in systems with access to pasture for at least part of the year, as is the case for 90% of cows in France, 95 to 100% in Ireland, 99% in Australia and New Zealand [13,14]; however, these systems differ in the management of fattening cattle in tropical systems. Additionally, some of the measures used for animal welfare assessment under confinement conditions (e.g., lameness score, social behavior), are relevant also for grazing cattle [15]; however, it is not feasible to transfer assessment protocols developed for intensive systems to grazing systems, because each system needs a specific protocol and the proposed indicators are not necessarily suitable, relevant, feasible or measurable under grazing conditions [15,16]. Therefore, protocols with an evidence-based approach are needed to assess animal welfare in extensive fattening systems in tropical climates [2].

Beef production systems may be broadly classified as extensive, including rangeland and pastoral, agropastoral, mixed farming, and intensive. Pasture-based or forage-based production systems predominate in the main beef-producing countries such as the USA (West), Brazil, Argentina, Australia, New Zealand, Canada, and Uruguay, in some European countries such as France, United Kingdom and Ireland, and in sub-Saharan Africa, depending on available feed resources, the environment, market requirements and costs of production. Cattle in pasture-based systems are subject to high levels of environmental variation to which specific genotypes are better suited. They include grazing and rangeland production within beef only or mixed livestock and farming systems [17]. Colombian cattle ranching is distributed among 633,841 farms (n = 29,301,392 animals) [18], contributes 6% of the national gross domestic product and generates 810,000 direct jobs. Fattening cattle represents 20% of the inventory and is managed under grazing conditions [19]. Currently, the National Animal Welfare Council and the Technical Committee on Production Animals were formed [18], which, jointly with producers’ unions, academia, governmental entities, and citizen participation, prepared the manual and methodology for the evaluation of animal welfare conditions in cattle and buffalo farms [20]. However, the measures used in the methodology and guide have not been tested to establish their applicability in traditional extensive fattening systems, so that they can become a tool for producers to improve animal welfare conditions, monitor their production systems, and evaluate progress over time. This protocol was designed by the Colombian health authority in conjunction with producers, academics, and professionals in the area, to have a regulatory tool that establishes the minimum guidelines for animal welfare on farms, through the implementation of actions that allow its adoption and the commitment of the actors that participate in the bovine meat chain. The objective of this study was to test the feasibility of the proposed measures and methodology for the evaluation of animal welfare in fattening cattle under pasture conditions and to propose other indicators or methodologies to complement the evaluation. The proposed protocol can be an evaluation reference for extensive and pasture-based systems in other countries with similar characteristics [18].

## 2. Materials and Methods

### 2.1. Ethical Consideration

The study was carried out under commercial farm conditions and the researchers participated in the process solely as observers. All procedures related to the use and care of the animals strictly followed the Colombian regulation norm, Resolution 001634-2010, as stated by the Colombian Agricultural Institute [21]. Permission to conduct the study was approved by the Ethics Committee for Animal Experimentation (Act 30/12/2021, activities with minimal risk) at the University of Caldas. Farmers were fully informed about the purpose of the study, and they read/listened and signed an informed consent form, and authorization to allow us to use the data and the pictures taken on the farms.

### 2.2. Development of the Evaluation Protocol

The methodology for the assessment of animal welfare in cattle and buffalo (Version 1.0) was developed in its initial phase as an initiative of the Colombian Federation of Cattle Breeders and the National Livestock Fund (FEDEGAN-FNG) with the participation of an international expert in livestock animal welfare. The second phase was developed with the objective of socialization, adjustment, and initial validation of the protocol, through the implementation of “workshops with a participatory approach and knowledge management” [22], which took into account (a) the tacit knowledge acquired through life/work experiences and oral traditions; and (b) the explicit knowledge based on the scientific knowledge of validated protocols such as Welfare Quality^®^ [23]. Six national workshops were held, involving professionals and producers from the departments of Antioquia, Córdoba, Meta, Vichada, Guaviare, Arauca, Casanare, Atlántico, Magdalena, Cesar, Guajira, Bolívar, Sucre, Santander, Tolima, Huila, Cauca, Valle del Cauca, Caldas, Risaralda, Quindío, Boyacá, Cundinamarca and Nariño. In the third phase, the regulatory entity and a panel of experts in animal welfare evaluated and complemented the methodology for the assessment of animal welfare in cattle and buffalo species proposal for its implementation in Colombia [18].

### 2.3. Farms Selection

The study was conducted in 24 commercial pasture-based Zebu beef farms in the Magdalena Medio region in Colombia, South America, visited between April and July 2022 (Table 1). Herd sizes ranged from 20 to 1300 animals, with an average age and weight at slaughter of 2.6 ± 0.1 and 503.3 ± 8.4 kg, respectively. The selection of the farms was made taking into account the following inclusion criteria: (a) the management of cattle belonging to commercial Zebu cattle crosses; (b) the raising and fattening of the animals were carried out on the same farm under pasture/grazing conditions; (c) the slaughter of the animals was carried out in the two slaughterhouses in the region (category A, national consumption), to monitor the animals during the ante-mortem and post-mortem inspection to develop a second phase of the project, whose results will be disclosed in a second publication; and d) the producers’ voluntary involvement in the project.

### 2.4. Description of the Study Area

The animals were located in the Magdalena Medio region (low tropic), which corresponds to an extensive mid-Andean valley in central Colombia, formed by the Magdalena River and distributed in the departments of Antioquia, Bolivar, Boyacá, Cesar, Caldas, and Santander. This cattle-grazing area is considered promising for achieving livestock production in harmony with forests and wetlands [19] (Figure 1). Rotational grazing was carried out on improved grass pastures of *Brachiaria decumbens*, *Brachiaria humidicola*, *Dichanthium aristatum Benth*, and *Megathyrsus maximus*, among others. In addition, there were natural grasses, native legumes, and native trees (*Xylopia amazonica*, *Clathorotropis brachypetala*, *Lecythis* sp.). The beef cattle belonged to commercial Zebu cattle crosses and some producers were making F1 crosses with Bos taurus breeds. The entire production process was carried out on pasture. The calves were raised and fattened on nearby farms of the same owners or were marketed through auctions and livestock markets, to complete their production cycle on the buyers’ farms in the same region. Calves were kept with their mothers until they were nine months old and reached an average weight of 180 kg; the calves were then placed in rearing (240–260 kg) and fattening lots until they reached the average market weight (503.3 ± 8.4 kg), and were kept in the same social groups until they were transported and slaughtered [19]. At all stages of production, the animals are supplemented with formulated mineralized salt in accordance with their nutritional needs.

### 2.5. Field Application of the Methodology

One of the article’s authors, who has more than 12 years of PhD education and training in bovine animal welfare assessment, applied the methodology in the company of the master’s program student. Table 2 presents an overview of the measures (indicators) proposed in the original methodology (Version 1.0) organized into four domains (good nutrition, appropriate environment, good health, and appropriate behavior). Observations began early in the morning, starting in the pastures to evaluate animal-based indicators in cattle with a body weight of more than 350 kg that had already adapted to the fattening conditions, to avoid measurement biases. The resource-based indicators (included in the methodology) were also evaluated, and the visit ended with evaluating the management-based indicators.

A visit was made on horseback (minimum two pastures per farm) to evaluate by direct observation the cattle in the pastures and the environmental conditions, forage supply, access to water and shade, and general condition of the pastures and fences. A proportional random sampling was performed according to the proposed methodology and the total number of cattle in the herd, to evaluate animal-based measures (body condition, presence of lesions and inflammations, biting flies, ticks, worms, and lameness), taking into account the following criteria: (a) in a farm with <10 animals, the whole group was assessed; (b) 11–20 animals, 10 animals were selected; (c) 20–99 animals, 20 animals were selected; (d) 100–499, 30 cattle were sampled; (e) 500–999, 40 animals were selected; and (f) >1000 animals, 5% of the population [24].

To evaluate the reaction of the animals to humans, the evaluator observed the behavior of each lot of the cattle for 1 min when the handler entered the pasture on horseback and approached within one meter of the batch, with the evaluator remaining 2 m from the group in a static posture, without interfering with the evaluation. Behavior was evaluated subjectively and was determined according to the reaction and movement of the animals. Two classes of behavior were assigned to the batches: (a) calm, quiet, and still animals, with little or no resistance to being approached, and (b) excitable: constant and energetic movements, with attempted escape, very agitated and frightened and, in some cases, with uncontrolled movements [25]. By this means, the proportion of animals with calm or excitable behavior was established. The cattle handler was then asked to interact with a group of cattle in the pastures (the cattle handler rode through them without tactile contact), to evaluate the human–animal interaction. The reaction of the animals was generally recorded as positive, negative, or neutral [24]. It was considered positive when the batch of animals willingly sought voluntary engagement and spatial proximity, as well as signs of anticipation, pleasure, or relaxation (the posture of the head, ears, and a relaxed body, showed interest in the handler). It was considered neutral when the animals were calm during the interaction, with the head in a normal position, and non-erect ears, while at the same time looking at the handler and continuing to ruminate. A negative reaction was when the response of the animals was avoidance, flight, or vigilance [26]. Figure 2 presents an overview of the typical grazing conditions in the Colombian Magdalena Medio, the crossbreeds of Zebu cattle in the area, and the interaction of the cattle with the handler.

The information of the management-based indicators was obtained through documentary evaluation (review of procedures, and records), and a structured interview guided by means of a questionnaire with the owner or administrator of the farm to evaluate the health and general management of the cattle. These factors, among others, were addressed in the questionnaire: water, feed, and mineralized salt supply; routine management practices; painful procedures (dehorning, castration, and hoof trimming), personnel expertise, the use of analgesia, anesthesia, age of the cattle when the procedure was performed; vaccination; diseases observed in the animals; mortality (frequent causes); culling (% and causes); average weight and age at slaughter; animals that required special care (%, causes); and and handling of sanitary and production records.

### 2.6. Assessing the Feasibility of the Proposed Welfare Measures

Software Stata Version 13.0 (College Station, TX, USA) was used for all the statistical analyses. A descriptive analysis of the information obtained from the 24 farms was performed and Spearman’s coefficient was used to identify measures with a strong association (ρ ≥ 0.8), to select measures that could be evaluated employing a common indicator.

After completing the farm visits, an evaluation of the indicators proposed in the protocol was carried out with the participation of two international experts in bovine animal welfare and by a panel composed of the authors of the article, two veterinarians with a doctoral/Ph.D—training in animal welfare, and four representatives from producer organizations, and the sanitary authority. Once the analysis was completed, version 2 of the protocol was elaborated with the adjustments proposed by the panel [27].

The evaluation of the feasibility of the measurements proposed in the first version of the protocol considered the following criteria: (a) ease of recording by a single assessor in all field conditions, (b) non-invasive measurements for the animals, (c) measurements that did not require additional animal handling, (d) applicability of the measurement in pasture-based production systems, (e) time and space constraints, and (f) measurements that needed specialized assessments [2]. Additionally, a comprehensive review of the scientific literature on animal health and welfare, existing recommendations, and legal requirements was made to select the most widely used and feasible measures to be applied in these systems.

The authors divided the protocol indicators into three groups, according to the feasibility of their application, following the guidelines proposed by Kaurivi et al. [16]:

(a) Indicators that remained unchanged in the final protocol;

(b) Indicators that were considered necessary and adequate to be maintained in an adjusted form in the final protocol;

(c) Indicators that were removed from the protocol.

## 3. Results

### 3.1. Field Application of the Methodology

A total of 24 farms and 788 fattening cattle were evaluated in the pastures to evaluate the feasibility of the animal-based measures. The complete evaluation on each farm took between 2 and 2.5 h depending on the number of animals and the distance to reach the pastures. Accurate sampling of cattle in each pasture was difficult because the animals were not subjected to any containment procedures. Water treatment was not carried out on 95.8% of the farms (n = 23), nutritional supplementation of cattle with mineralized salt was carried out on 100% of the farms and with hay on only 20.83% (n = 5); animals had clean and safe resting places on 91.67% (n = 22) of the farms and 79% (n = 19) of the pastures had trees and natural free-range access.

Twenty-five percent of the cattle (n = 197) had a body condition between 3 and 3.5. A total of 79.2% (n = 19) of the farms performed animal branding with a hot iron, without anesthesia and analgesia; castration of animals with anesthesia, but without analgesia was performed in 87.5% (n = 21) of the farms and the remaining 12.5% (n = 3) did not perform this management practice. Dehorning was not performed on 58.3% (n = 14) of the farms and the most used method was the application of a topical product. The evaluation of the presence of lameness in the pastures was difficult in some farms, due to the topographical characteristics, but when interviewing the cattle handlers and/or owners about this aspect, 29.17% (n = 7) expressed that it was a problem that occurred at certain times of the year in the region, due to the flooding characteristics of the pastures. No bovine stereotypies were observed, but some affiliative behaviors were observed, which were not included in the protocol. The difficulty was encountered in quantitatively assessing the level of ectoparasite infestation of the animals and the presence of lesions. Only one respondent reported the mortality and cull of a bovine in the last 12 months.

None of the handlers had certified animal welfare training. Registration of treatment of cattle by a veterinarian and current health plans were not routinely carried out on the farms. A total of 100% of the evaluated groups of animals exhibited calm behavior when the handler interacted with them in the pasture and when approaching the group, the predominant response was a positive interaction (79.2%). The proportion of animals requiring special attention was low (n = 1).

### 3.2. Feasibility of the Measures Assessed

The panel made the selection of the variables to be included in version 2 of the protocol considering the criteria established in the methodology and the measures proposed and validated in the scientific literature. Of the 30 pre-selected measurements, 28 were finally selected and two new were proposed. Unsuitable criteria and measures excluded as not applicable for an animal welfare assessment, because they are part of good primary production practices. Table 3, Table 4 and Table 5 describe the reasons why each indicator was classified in each of the proposed categories (included in the protocol without modification, included with adjustments, not included).

## 4. Discussion

In the United Kingdom and Germany, the beef industry is at the forefront in adopting welfare outcome measures as part of its farm assurance scheme [28,29]. The Colombian Federation of Cattle Breeders-FEDEGAN-proposed the evaluation protocol under validation in this study. These on-farm assurance program initiatives indicate the commitment of producers to manage their animals according to animal welfare standards, which then allows them to access certain markets [30], sell their products to more demanding retailers [31], change management routines [32], monitor and evaluate changes in practice, target interventions based on results [33], and comply with the requirements of sanitary legislation [34]. These evaluation schemes favor active participation and a long-term commitment to improvement by producers and greater adherence to animal welfare audit protocols [29,35].

### 4.1. Field Application of the Methodology

The proposed methodology lacks measures that evaluate the quality of life of animals, defined as “subjective and dynamic evaluation by the individual of its circumstances (internal and external) and the extent to which these meet its expectations” [36,37]. Quality of life assessment has been used in canine and feline research but little has been applied in farm livestock species [38]. Its approach is challenging in pasture-based systems, however, because the concept of animal welfare is complex and multifaceted, and animals do not respond in the same way to environmental stressors, management and their inherent characteristics, and how this has implications on animals’ physical and emotional/mental state [39]. However, a study comparing the welfare of housed and grazing cattle using qualitative behavioral assessment (QBA) reported grazing as a better management system in terms of welfare, mainly due to a higher prevalence of positive behavior [40].

### 4.2. Feasibility of the Measures Assessed

Grazing and foraging systems for beef production vary widely depending on environmental, animal, and economic factors and their interactions [17]. Therefore, the methodology proposed is focused on critical aspects that could impact cattle welfare regardless of the aforementioned conditions, concentrating on suffering and needs as indicated by the Five Freedoms paradigm and giving animals a life worth living [2]. We will discuss those measurements organized into four domains (good nutrition, appropriate environment, good health, and appropriate behavior).

#### 4.2.1. Good Nutrition

This methodology evaluates the criteria of “absence of hunger and thirst” through direct observation of the accessibility, availability, quality and quantity of forage and water in the pastures. In systems based on pastures, particular attention must be paid to water provision. Water provision and animal welfare are closely connected, and climate change might further compromise animal welfare, especially in geographical areas affected by droughts [41]. In Colombia and other countries with a wealth of natural water, there is an abundant supply of water sources such as rivers, natural wells, and springs, which are used for livestock consumption. However, despite their ecological importance, cattle ranchers are transforming these ecosystems and have suggested the adoption of water collection, storage, and conduction systems, so that each pasture has a drinking trough, to prevent cattle from entering riverbeds and thus preserve and recover the riverside areas, marshes, wetlands, rivers, micro basins and conserve wildlife and fish [19,42]. The inclusion of this indicator of accessibility to food and water, as described in the original methodology, is suggested because it is supported by scientific evidence [43,44].

Other authors have proposed relevant indicators such as counting the number of working water points (especially in natural water troughs), the flow rate, length of water troughs, classification of water troughs as safe or unsafe (slip hazards, presence of risks, bearing capacity of the soil, etc.), observation of the competitive behavior of cattle in front of water sources, and the cleanliness of the water (absence of odors and strange colors) [16,43,45]. Other alternatives are (a) evaluation of the distance traveled by the animals to access water, because it has been suggested that if the water supply sources are located more than 250 m away, cattle decrease their water consumption [15], but animals in tropical conditions have a permanent supply of green forage, which decreases fresh water consumption by animals; (b) how access to water is managed during grazing [43]; and (c) feeding strategies to check that the pasture provides sufficient nutrients during grazing [15].

In the interviews conducted for the validation of the protocol, the existence of water treatment systems and supplementation based on nutritional analysis instead of pasture were inquired about, which are infrequent practices among producers (<5%). On the other hand, there were no laboratories in the area in which to perform the tests and there are other indicators such as body condition and daily weight gain of the animals that are more efficient. However, it is important to keep in mind that strategic nutritional supplementation can contribute to reducing deficiencies in the quantity and quality of feed based on pasture or forage and contribute to obtaining higher productive yields [17].

Body condition scoring is an effective measure of medium-term energy balance and is proposed as a unique indicator to assess nutritional performance on dairy farms in New Zealand [44] and considered feasible in this methodology with animal sampling. This procedure was performed on horseback to obtain a close proximity to the cattle and is considered efficient by other authors [15]. Optionally, the measurement of body condition could be performed in the pen during the vaccination process of the animals or by monitoring the batches in the slaughterhouses, through the evaluation of hot and cold carcass yield, as has been proposed in swine in Colombia [45].

#### 4.2.2. Good Environment

One of the most complex aspects to evaluate quantitatively is the comfort of fattening cattle, because grazing conditions are variable, do not remain stable over time, and can fluctuate due to weather conditions, animal management, pasture quality, and pasture rotation, among others. These aspects, in turn, induce variations in resting areas, feed availability, distances to be covered, forage quality and quantity, soil quality and susceptibility to heat stress, among others [44,46]. One of the most complex aspects to evaluate quantitatively is the comfort of fattening cattle, because grazing conditions are variable, do not remain stable over time, and can fluctuate due to weather conditions, animal management, pasture quality, and pasture rotation, among others; these aspects, in turn, induce variations in resting areas, feed availability, distances to be covered, forage quality and quantity, soil quality and susceptibility to heat stress, among others. Colombia is developing an industry technical standard that establishes environmental requirements for the livestock industry and the creation of an environmental seal in response to market demands for the adoption of sustainable practices through the management of good irrigation practices, planting, land management, waste management, good livestock practices, and social responsibility with employees [47]. This policy is based on the establishment of silvopasture systems and forest conservation on cattle ranches as a strategy and opportunity for environmental offsetting [19].

Assessment of thermal stress has been performed by observing respiration patterns or through temperature measurement. However, these measurements do not appear appropriate for beef cattle systems where animal restraint possibilities are few compared to dairy systems [48]. In this study, thermal comfort was assessed by direct observation of the resources available to help cattle cope with heat or cold stress (e.g., the presence of trees in the pasture, and silvopasture systems), which is considered valid in pasture-based systems [44] and is valuable to producers as it provides protection against extreme climates and contributes to wildlife conservation [49]. Other indicators have been suggested such as (a) the evaluation of the cleanliness of the animals’ hind quarters and the percentage of dirty animals [50], but these indicators are not very applicable when the animals are free in the pastures, due to the difficulty of observation; however, it is considered valuable because the presence of mud or manure, are risk factors for the presentation of lameness [11,51]; (b) the use of sensors to evaluate animal behavior during long-term grazing (an option that would be valid, if the evaluation were performed for research purposes); (c) evaluation of resting behavior (animals lying outside or inside the resting or shaded area [45]; and (d) measurement of clinical signs of heat stress (such as panting) [7]. However, a single evaluation is not considered representative of the entire pasture [3] and requires additional time for evaluation.

The proposed methodology for beef cattle evaluated comfort around a resting place through the observation of pasture conditions (presence of floodable areas). In this particular case, the Magdalena Medio region has 70% of hilly areas and the remaining 30% corresponds to floodable forests of the Magdalena River, rich in native legumes, natural succession trees, and a high biodiversity of wild species [19,52]. The implementation of this indicator is suggested because it is easy to measure and we followed up on the foot lesions of animals from the evaluated farms during post-mortem inspection in two slaughterhouses and found a high frequency of lesions related to pasture moisture conditions, such as heel erosion and abnormal claw shapes (asymmetric claws and corkscrew claws), results that coincide with those described by Bautista-Fernández et al. [53] in Mexico and by Moreira et al. [51] in Brazil.

These lesions cause pain, decreased feed intake, and significant economic losses [54]. In addition, this indicator is associated with the thermal comfort of the animals [55], the presence of lameness, and other abnormalities of the hooves. This finding could lead to the strengthening of sanitary programs for the monitoring and treatment of heel erosion and hoof problems, an activity that was routinely carried out in the region, according to the information provided by the interviewees.

The methodology proposed the evaluation of the general resting conditions of the animals in clean and dry places an aspect that is easy to measure because during the visit the animals can be observed together resting in these areas. This evaluation is very important, because excessive mud is a problem that generates chronic stress and affects health, feed conversion and weight gain [56]. On the other hand, it has been suggested to use, as a measure of thermal comfort, adaptations made by the owners to provide greater comfort to the animals (environmental enrichment), but we considered that the environmental enrichment strategies are naturally included in the pasture environment [49].

#### 4.2.3. Good Health

Veterinarians, farmers, and cattle handlers have a responsibility to promote principles for pain control in animals under their care for ethical and animal welfare reasons. Unfortunately, in farm animals, pain has traditionally been overlooked as they are assumed to be less sensitive than pets [57], the use of pain-mitigating products is discouraged due to the costs of the procedure (labor, medications, specialized personnel, and time required), attitude, negative beliefs [58] and lack of empathy towards the animals, amongst others [59]. However, castration, branding, and horn removal, regardless of the technique used, generate a pain-inducing response; however, these procedures are usually performed without drug administration [60].

Information related to the management of painful procedures was obtained by means of an interview, as proposed in the Welfare Quality^®^ protocol, a validated and easy-to-apply methodology [23]. In this study, it was observed that cattle castration and dehorning are common livestock practices that are frequently performed without pain mitigation, results that are consistent with other studies [61]. It has been reported that the use of analgesics, anesthetics, and anti-inflammatory drugs is a more frequent practice in adult cattle than in suckling and newborn calves, according to studies conducted in Brazil [60]. It is suggested the implementation of promotion and continuing education programs by veterinarians, producer organizations, and state institutions, among others, to promote pain mitigation practices, as well as the development of research to evaluate their efficacy in reducing pain-induced distress and post-treatment responses of cattle (e.g., inflation after surgical dehorning) [60].

Mortality was evaluated as the percentage of animals that died from all causes (disease, accident, no specific cause) during the last year, as indicated by the Welfare Quality^®^ protocol [23]. Respondents reported percentages below 1% gross mortality, coinciding with results obtained in studies conducted in pasture-based dairy systems [7,62]. It is suggested that the indicator called “complementary care”, which corresponds to the % of sick or injured cattle that do not receive timely treatment and attention (separation of the herd, provision of soft bedding, access to water and feed, and treatment), can be retained without modification because during the visits these practices were observed and cattle handlers daily check the health conditions of the animals and implement the relevant sanitary measures.

If an assessment protocol is to be widely used, it must include individual assessments that are practical to measure within the system being assessed, and it must be feasible within a reasonable time frame [44]. The latter is particularly important in a pasture-based system, as many animal-based assessments can only be measured while milking or performing a medical procedure on beef cattle (vaccination and drug administration), as this is the only time when they can be closely and systematically observed, an aspect that limits their implementation. In this study, the evaluation of animal-based measures (ectoparasite infestation, and the presence of inflammations and lesions) was made difficult by the dispersion of the animals in the pasture and the difficulty of establishing the level of ectoparasite infestation by detailed observation of the animals in areas that are difficult to access (behind the ears, groin, base of the tail and udder/testicles) as proposed by the evaluation methodology for both dairy and beef cattle [63]. The second version of the methodology contemplates the transfer of a sample of the animals in pastures to pens to carry out the evaluation of these measures, which we consider inappropriate because it would cause additional stress to the animals. In New Zealand, it has been recommended to obtain information through guided interviews on the prevalence of problems caused by ectoparasites and control strategies (rotational grazing, integrated parasite control, etc.) [16], an aspect that we suggest as feasible to implement. However, we recommend assessing the presence or absence of ectoparasite infestation in the pastures and supplementing the information through a focused interview.

Several authors indicate that lameness is more frequent in feedlots [2], an aspect that has been widely demonstrated; however, the observation of lameness in pasture-based dairy systems [11] was common. Lameness can also be evaluated during a visit to the pasture on the day of the audit, by having the handler move the cattle.

Other specific risks for fattening cattle on pasture that could be inquired by interview is the presence of photosensitized animals (frequent in some tropical regions) [64], ingestion of toxic plants [65], gastrointestinal parasites (e.g., *Ostertagia ostertagi*) and the strategies used for their control [62] and hemiparasites (*Anaplasma marginale*, *Babesia bigemina*, *B. bovis*, *Tryapanosoma* sp.) [66,67].

#### 4.2.4. Appropriate Behavior

Grazing is part of the natural behavior of cattle, which is reflected in greater social cohesion and permanent affiliative interactions, which reduce aggression, stereotypies, and stress in cattle, compared to those kept in confinement [10,62]. Pastures provide cattle with greater opportunities for exercise, access to varied feed sources, the ability to select feed according to their preferences, and low competition for resources [10,68].

This study evaluated the quality of human–animal interaction by observing in the pasture the positive behaviors of beef cattle, during interaction with handlers, as has been proposed in dairy cattle on pasture [44,50,69]. Most behavior data are collected by direct assessment for an unfamiliar evaluator. However, according to Hernandez et al. [50], approaching animals in extensive systems may be difficult and sometimes not very informative as cattle bred in large groups in extensive systems, like as systems under pastures, may avoid the human touch, even if not necessarily afraid of it. Additionally, the feasibility of direct assessment for behavioral observations is often low in pastures, because it needs time-consuming indicators, requires many trained evaluators and, furthermore, information provided about inter-observer reliability is not always sufficient [70,71]. This study inquired about the practices used by handlers in order to reduce animal fear responses and improve human–animal interaction during routine handling, an indicator that is considered valid, as significant correlations have been revealed between the practices reported by handlers and behavioral responses of beef cattle [72].

In cases where the objective of the evaluation of human–animal interaction on farms is the improvement in management, decrease in the risk of occupational accidents and promotion of the welfare of beef cattle, we suggest the evaluation of the quality of human–livestock interaction with the measures proposed in this methodology. Likewise, studies conducted in beef herds in French farms successfully validated the assessment of human–animal interaction by means of the behavioral test of animals’ reactions to humans (an avoidance test) and found a significant association with the practices reported by cattle farmers through a semi-structured interview based on three aspects: (a) the general description of farm and herd management practices (grazing period, and discarding and selling animals), (b) organization of work with cattle (frequency of herd monitoring, hoof trimming, feeding organization, type of herd monitoring in pastures) and (c) the handler’s relationship with cattle (number of accidents with cattle, value of having a good farmer–cattle relationship, methods that facilitate cattle handling) [72].

The evaluation of the presence of stereotypies in this protocol was considered not very applicable to evaluate in grazing cattle because its estimation requires the elaboration of a previous ethogram and a prolonged observation process and these are more useful to evaluate in stabled cattle [73]. On the other hand, it was considered relevant to evaluate the proportion of personnel with certified training in animal welfare and good management practices, because it is a requirement of the Colombian sanitary legislation directed to all animal handlers in logistic chains (farm, transport, livestock markets) [74]; in addition, studies conducted in Brazil in pasture fattening cattle farms found that training cattle handlers through an effective, practical and periodic strategy promotes positive human–animal interactions [75], improving the quality of life of both handlers and livestock [71,76].

## 5. Conclusions

The proposed methodology for the evaluation of fattening Zebu cattle under pasture conditions included animal-, resource-, and management-based evaluations, along with record-related evaluations. Although it is considered ideal for protocols to be based primarily on animal-based measurements, which allow for the estimation of actual welfare status in animal behavior, health, and body condition, resource-based and management-based measures are based on science and expert experience, making them feasible to implement. Despite the lack of representativeness of the evaluated farms of all existing types of fattening cattle on pasture, the proposed methodology yielded valid information to propose it as feasible, simple, and representative of animal welfare conditions in these systems. The proposed methodology is expected to be suitable for use at the farm level for comparative evaluation, allowing the monitoring of the progress of each farm and the certification of standards required by sanitary legislation. However, the next step is to evaluate the reliability of the measures, especially those that assess the repeatability of the results by assessing inter-observer reliability (which refers to the probability that two different people will produce the same results) and test–retest reliability (which refers to the probability that the same results will be obtained if the test is repeated).

Likewise, it would be valuable to include measures to assess the quality of life of animals such as the QBA, which integrates information on the body language of animals and how an individual animal interacts with the environment, to assess individual variation in livestock behavior and human–animal interaction.

Implementation of pain mitigation techniques for practices considered painful for livestock is deemed an urgent need from an ethical and animal welfare point of view. This would be achieved through continuing education programs provided at the beginning of veterinarians’ professional training, strengthening the attitude and empathy of farmers and professionals towards livestock, as well as the development of studies to identify the best medication and application of protocols, among other aspects.

## Figures and Tables

**Figure 1 animals-13-03659-f001:**
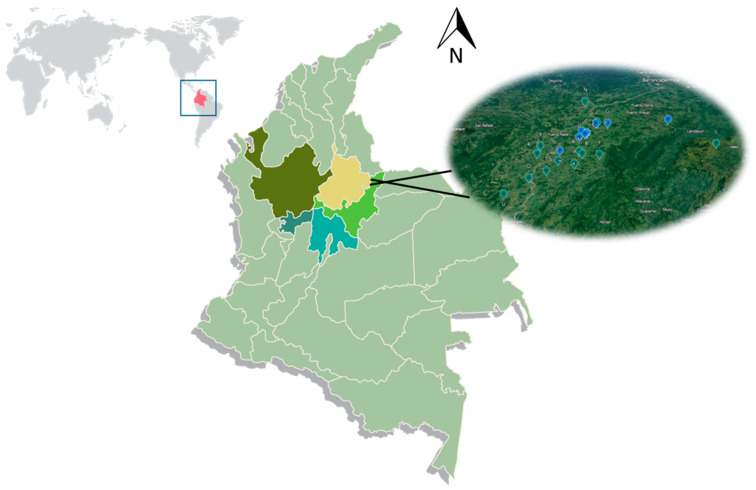
Geographic area evaluated in the study (Magdalena Medio, Colombia, South America).

**Figure 2 animals-13-03659-f002:**
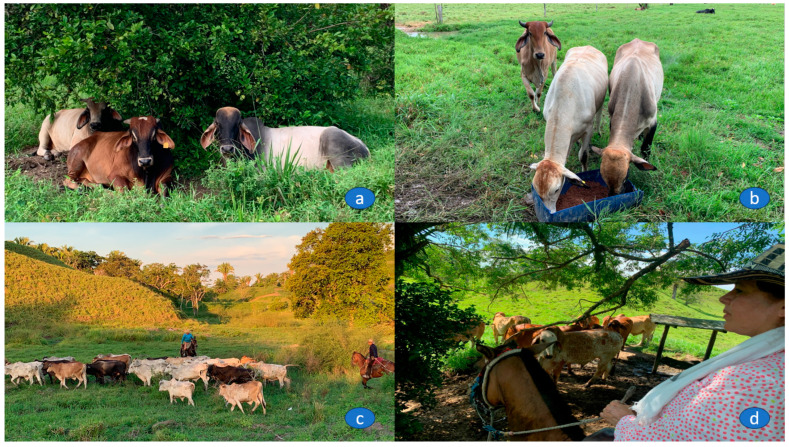
Grazing conditions in the middle Colombian Magdalena. (**a**) Thermal comfort. (**b**) Supply of mineralized salt. (**c**) Reaction of animals to humans. (**d**) Interaction with humans.

**Table 1 animals-13-03659-t001:** Description of farms evaluated in the Magdalena Medio regions (Colombia, South America).

Farm	Altitude	Livestock Numbers	Animals Evaluated	Farm Size (ha)	Welfare Score	Category
1	6°9′26″ N	535	40	333	44	High
2	6°7′58″ N	1100	55	600	48	High
3	6°7′58″ N	1100	55	600	59	High
4	6°7′58″ N	500	40	600	61	High
5	6°16′4″ N	1300	65	750	53	High
6	6°7′58″ N	1100	55	600	60	High
7	6°15′32″ N	580	40	370	58	High
8	5°56′32″ N	620	40	375	58	High
9	6°29′37″ N	1150	58	523	68	High
10	6°15′20″ N	398	30	382	59	High
11	5°55′34″ N	165	30	1200	57	High
12	5°59′28″ N	36	20	100	61	High
13	5°56′4″ N	1200	60	1500	58	High
14	6°36′10″ N	103	30	1200	70	Excellent
15	5°42′32″ N	76	20	190	62	High
16	5°58′34″ N	39	20	240	59	High
17	5°29′15″ N	20	10	300	56	High
18	5°54′41″ N	20	10	30	52	High
19	5°27′41″ N	24	10	13	57	High
20	5°45′45″ N	73	20	2250	50	High
21	5°36′11″ N	47	20	410	63	High
22	5°50′55″ N	53	20	100	53	High
23	5°59′16″ N	76	20	170	57	High
24	5°54′15″ N	47	20	119	58	High

**Table 2 animals-13-03659-t002:** General description of measure assessment in the Colombia beef farms in a pasture-based-system protocol (Version 1.0).

Principle	Welfare Criteria	Animal Welfare Measure/Indicator	Method of Assessment
Good nutrition	Absence of hunger	Access and availability of grass	Direct observation
		Quality and quantity of grass	Direct observation
		Supplementation based on nutritional analysis	Documentary record
		Fedd storage	Direct observation
		Body condition score	Animal-based indicator (animal sampling)
	Absence of thirst	Access of water Availability of water in drinkers and natural sources	Interview and direct observation
		Ad libitum	
		Water treatment/physicochemical and microbiological analysis	
Appropriate environment	Thermal comfort	Subjective assessment of shade in the paddocks	Interview/direct observation
		General condition of the paddocks	
		General condition of the facilities and fences	
		Adaptations that provide comfort to animals (grooming object, draft protection, shaded feeders and drinkers)	
	Comfort around resting place	Animal rest place (clean and dry)	
Good health	Painful procedures	Ear tagging/disbudding/castration	Interview
		Specify age at painful procedures	
		Procedure and with/without use of analgesia and or anesthetic	
		Staff expertise	
	Absence of disease	Abrasions/swelling/hairless	Animal-based indicators (animal sampling)
		Presence of biting flies	
		Presence of *Dermatobia hominis*	
		Presence of ticks	
		Presence of worms	
		Lameness	
		Animals requiring complementary care	Calculation/interview
		Mortality rate	
		Culling rate	
	Use of veterinary medicinal products	Evaluation of the drug storage area and good drug management practices	Direct observation/records
	Procedures and documentary records	Registration of treatment of cattle by a veterinarian and current health plan	
Appropriate Behavior	Human–animal interaction	Animal reaction to human	Animal-based indicators (animal sampling)
		Human–animal interaction	
		Stereotypes	
		Knowledge and training in animal welfare	Interview—% people with certified training

**Table 3 animals-13-03659-t003:** Measures assessed as feasible for inclusion in the final methodology without change.

Principle	Welfare Criteria	Welfare Measures/Indicator	Method of Assessment
Good nutrition	Absence of hunger	Access and availability of grass	Subjective assessment of grass in the paddocks (type and availability during the year) as enough or insufficient. Interview and direct observation
		Body condition score (thin animals)	% thin animals in the herd, based on score ≤ 2.5 on 1–5 scales Categorical scale according to the proportion of animals with score ≤ 2.5 to assign total score (0–8).
	Absence of thirst	Access to water in drinkers and natural sources	Subjective assessment of availability of natural water sources/drinkers as enough or insufficient
		Ad libitum/restriction of water	Interview/direct observation as enough or insufficient.
Appropriate environment	Thermal comfort	Shade and adaptations that provide comfort to animals (trees)	Subjective assessment of shade in the paddocks (presence of trees, shrubs, galleys).
		Adaptations that provide comfort to animals (environmental enrichment)	Direct observation
	Comfort around resting place	Animal rest place (clean and dry)	Subjective assessment of the potential resting places (animals probably stay together)
Good health	Presence of hazards	Fence status	Subjective assessment of fence condition in the visited pastures (intact, free from sharp elements or any other conditions that may cause harm or injury to the animals).
	Absence of pain from management procedures	Ear tagging, disbudding/castration	Record age of the animal, staff expertise and use of local analgesia and anesthetic during questionnaire-guided interview.
	Absence of disease and pain	Lameness (animal based-indicator)	At pasture/Subjective assessment % of cattle with uneven weight-bearing on a limb that is immediately identifiable and/or obviously shortened stride. Categorical scale: No lameness (0): normal displacement and poise Mild lameness (1): abnormality in displacement or posture Severe lameness (2): arching of the back. % cattle with severe lameness (≤5%, >5%, ≤10%, >10%)
		Mortality rate (%)	Interview/registers/calculation % Numbers of accidental deaths and deaths/slaughter (either on-farm or sent off-farm) due to disease were combined. (Excellent: ≤2%, High: 2.1–3%, Medium: 3.1–5%, Low > 5% or no records)
		Cull rate (%)	% culling/records/calculation Criteria: culling of the herd through a planned decision, age, slaughter on the farm, euthanasia procedure performed by a veterinarian, humanitarian slaughter (≤15%/>15%)
		Complementary Care	Animal-based indicator (sampling) % sick or injured animals not receiving timely treatment and care (herd separation, provision of soft beds, access to water and food, treatment) 1% Excellent 2% High 2.1–5% Medium >5% Low
Appropriate Behavior	Human–animal interaction	Reaction of animals to humans (Calm/excitable)	Animal sampling (animal-based indicator) Subjective assessment of beef cattle behavior when the rider enters the environment where the animals are located (reaction and movement of the animals) Categories: Calm–quiet (static animals, with little or no resistance to being approached) Excitable (constant, vigorous movement, attempting to escape, very agitated and frightened) [25]
		Interaction with humans (positive/negative/neutral)	Animal sampling (animal-based indicator) Subjective assessment of the animal’s orientation response to the handler Categories [26]: Positive: the posture of the head, ears and body relaxed, shows interest in the handler. Negative: vigilance, avoidance, flight. Neutral: Head in normal position and ears upright while looking at handler, continuous rumination.
	Knowledge and training	Formal training in animal welfare	Interview—% people with certified training in animal welfare (y/n, 100%, <50%, ≥50%)

**Table 4 animals-13-03659-t004:** Welfare measures included in the methodology after adjustments, including the rationale for change and the changes that were made.

Principle	Welfare Criteria	Welfare Measures	Method of Assessment	Reason for Difficulty	Adjustment of Measures
Good nutrition	Absence of hunger	Supplementation based on nutritional analysis	Interview and records	Some small producers do not perform this practice routinely because they do not have specialized laboratories in the area and the measurement of the animals’ body condition is a more effective measure.	Can be considered in systems that use strategic nutritional supplementation
	Absence of thirst	Water treatment and laboratory analysis (physiochemical and microbiological)	Interview and records	It is not a viable practice when working with natural sources. In many regions on the farms there is no supply of drinking water for humans and the water is obtained from wells	Can be considered in systems that use treated water
Appropriate environment	Comfort around resting	Hazards	General condition of the paddocks Direct observation	The protocol did not include categorical measurement scales and the presence of other hazards reported in other extensive production systems	Subjective evaluation of paddocks, including the identification of flood-prone areas and potential hazards within pastures such as steep hills, cliffs, gullies, and sinkholes. Also, noting the presence of hazardous objects or debris. Categorical scale: presence y/n
Good health	Absence of disease and pain	Abrasions/swelling/hairless (y/n)	Animal-based indicator Direct observation in the standing animal from a distance of no more than 2 m, of the presence of areas of alopecia or scars greater than 2 cm, dividing the animal into three zones: (a) head and neck; (b) body—trunk; (c) front and rear limbs	Difficulty in observing some areas in detail	Evaluation of the presence or absence of lesions to establish the percentage of affected animals.
	Presence of ectoparasites	Presence of biting flies (*Tabanus Stomoxys calcitrans*)	Animal based indicators/animal sampling Presence of biting flies on head, back, belly and legs (and/n, 50 insects)	Difficulty in observing some areas in detail and counting the number of insects present	Evaluation of the presence or absence of ectoparasites to establish the percentage of infested animals and follow-up the measurement with an interview on the prevalence of ectoparasites, problems caused, and control strategies
		*Dermatobia hominis/*ticks/worms	*Dermatobia hominis* (y/n, ≥5 insects) % of animals with ticks % of animals with myiasis % of animals with worms % of animals with horseflies	Difficulty in observing some areas in detail and counting the number of insects present	
		Presence of ticks	Direct observation of infestation on ears, groin, base of tail and udder (y/n, presence at least two of them)		
		Infestation by fly larvae	Direct observation of the infestation (y/n, presence at least one of them)		
	Disease history	Hemoparasites (*Babesia* sp., *Anaplasma marginale, Trypanosoma* sp.)	% of animals with clinical signs, diagnosis and treatment for hemoparasites	During the interviews the producers and/or administrators reported frequent health problems due to blood parasite infestation	Interview on the prevalence of blood parasites, results of diagnostic tests, problems caused and control strategies
		Photodermatitis	% cattle who became ill/treated during the last 12 months by photosensitization	These measures were suggested	
		Euthanasia protocol	% cattle that died on the farm during or subjected to euthanasia the last 12 m		

**Table 5 animals-13-03659-t005:** Welfare measures removed from the protocol after feasibility testing on 25 beef farms.

Principle	Welfare Criteria	Welfare Measures/Indicator	Method of Assessment	Reason for Removal
Good health	Absence of disease and pain	Use of veterinary medicinal products	Direct observation/records Drug storage area assessment Registered with the competent entity. Fed storage conditions Validity Veterinary prescription	It is part of good primary production practices, and no report was found on its use to evaluate animal welfare.
		Procedures and documentary records	Direct observation/records Record of treatment of cattle by a veterinarian Written health plan signed with a veterinarian with current professional registration	

## Data Availability

Data will be made available on request.

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
