# Peer review of "Study of the Feasibility of Proposed Measures to Assess Animal Welfare for Zebu Beef Farms within Pasture-Based Systems under Tropical Conditions"

_animals, 2023, doi:10.3390/ani13233659_

Round 1

Reviewer 1 Report

Comments and Suggestions for Authors

Dear authors,

paper describes an interesting topic, namely animal welfare what is becoming more and more prevalent issue in animal keeping. By comparing animal welfare in farms shortcomings can be identified and improvements can be made to bring the farms on a higher animal welfare level. A lot of energy has clearly crept into the paper, but it is not yet ready for publication. There are still some key issues to be worked out:

1) M&M: including more subtitles and adding more information will give a clearer structure to the M&M.

2) As the paper is written, the impact of the 2 experts on the protocol is very high and the statistical analysis seems incidental. While the statistical analysis is the most objective tool. Adjust or justify this in the paper.

3) A good paper has an identical structure in the M&M, the results and the discussion. A clear structure helps make the paper easy to read. However, in your paper the structure in the M&M, the results and discussion differs strongly. So, adjust the structures of the M&M, the results and the discussion to make the structure more identical.

4) some specific suggestions are made:

Line 44: ‘…consumers are willing to pay more for…’: are they? Are they doing it effectively or do they saying it during surveys? It seems in a lot of studies that there is a difference between paying more and saying to pay more for animal welfare.

Line 83: remove the dot in the title.

Line 108: ‘Farm Sample’: in a small farms the sample rate is 30 till 50 percent; in large farms the sample rate is only 5 percent. How can you justify the difference in sample rate in small and large farms?

Line 109: Please add the procedure how the 24 farms were chosen. I don’t think that the farms candidate spontaneously; or did they? Were there more than 24 farms/candidates and did you made a selection? If there was, how did you made the selection?

Line 109: where there any restrictions or requirements as farm to be a candidate for the animal welfare assessment?

Line 129: ‘The animals were kept in groups without social mixing during the entire fattening process’: it is not clear when the fattening period starts and ends. Please add this information. Thus, before the fattening period the animals were mixed a few times or once?

Line 133: I don’t think the title is correct. Is it?

Line 134: ‘A trained veterinarian…’ Thus, 1 person visited all the farms? So, there can be an observer effect? How was the veterinarian trained? Was his or her training validated?

Line 162-164: ‘After completing the farm visits, a critical evaluation of…’: remove the word ‘critical’ because the word is superfluous, you must be always critical…

Line 162-164: ‘…was carried out with the participation of two international experts…’: Why 2 experts? Can you justify the number of experts? Should be 1 expert enough? Or should a panel be better? And what is the definition of an expert? When can you say he or her is an expert (what characterises the experts (he is trained, years of experience, he is a researcher,…)?

Line 170: I miss some information, for e.g. about the weather conditions, the time of observation (the start and stop time)… These parameters can have an impact on the results and hence on your protocol.

Line 171: ‘Statistical analysis’: how did you know that the sample size was sufficient for statistical analysis? Did you do a power analysis before the start of the assessment? If not, it is still interesting to do it.

Line 171: it is not clear why you used statistical analysis: to include or to exclude indicators? Clarify in the text.

Line 176-182: this is not information about statistical analysis. It is necessary information, but not under the title ‘2.5 Statistical analysis’.

Line 189: ‘Table 3’: in the column on the right: at the first welfare criteria you write ‘Subjective assessment…’ but not at the last welfare criteria (lameness, complementary care, human-animal interaction…). Are these not assessed subjectively? Is there a difference between the assessment of the (first and the last) criteria? If it is, what is the difference?

Line 193: ‘Table 4’: adjustments of measures: ‘interview on…’: gives an interview objective information to include in the protocol? A respondent can say anything during an interview, but is not always correct… Isn’t it better to exclude these parameters than include information that is not validated for its correctness?

Line 196-199: ‘…the difficulty of measuring them on all farms, animal welfare implications not very applicable to the fattening production system under pasture conditions, time and space limitations, measures requiring specialized evaluations and adjustments to place the indicator in a category, where it was more feasible to evaluate it’: are these valuable arguments to exclude them? Or are these excuses to exclude them? And why not exclude more indicators on this basis? In my opinion is a detailed argumentation to include or exclude an indicator strongly needed. 

Line 205: ‘Table 5’/discussion: I’m not agree with the argumentation of the removal of the indicators (column right). I don’t believe that important welfare criteria as hunger and thirst can be excluded from a valuable animal welfare assessment protocol. If there is enough grass on the pasture, you can suggest the animals are not hungry, like you wrote. If there is a natural well in the neighbourhood (distance between animals and well) you could also suggest that the animals are not thirsty, etc.

Line 219: ‘Discussion’: The discussion is about the justification for excluding certain animal welfare criteria, but should rather be about solutions to include certain criteria anyway. How can you talk about valuable and reliable animal welfare assessment protocol when a lot of parameters are excluded.

Comments on the Quality of English Language

The written text is OK, but here and there are some minors. 

Author Response

  • M&M: including more subtitles and adding more information will give a clearer structure to the M&M.

Response: It was changed as suggested (lines 103-260).

  • As the paper is written, the impact of the 2 experts on the protocol is very high and the statistical analysis seems incidental. While the statistical analysis is the most objective tool. Adjust or justify this in the paper.

Response: It was justified (see lines 240-245).

  • A good paper has an identical structure in the M&M, the results and the discussion. A clear structure helps make the paper easy to read. However, in your paper the structure in the M&M, the results and discussion differs strongly. So, adjust the structures of the M&M, the results and the discussion to make the structure more identical.

Response: It was changed as suggested (see M&M, results and discussion)

  • some specific suggestions are made:

Line 44: ‘…consumers are willing to pay more for…’: are they? Are they doing it effectively or do they saying it during surveys? It seems in a lot of studies that there is a difference between paying more and saying to pay more for animal welfare.

Line 83: remove the dot in the title.

Response: The information about consumers was corrected and we remove the dot (see lines 47-51)

Line 108: ‘Farm Sample’: in a small farms the sample rate is 30 till 50 percent; in large farms the sample rate is only 5 percent. How can you justify the difference in sample rate in small and large farms?

Response: These guidelines were established during the construction of the protocol in the six workshops developed in the main fattening cattle-producing areas of the country, as briefly explained in section 2.2 of the article. We added complementary information (lines 192-195)

Line 109: Please add the procedure how the 24 farms were chosen. I don’t think that the farms candidate spontaneously; or did they? Were there more than 24 farms/candidates and did you made a selection? If there was, how did you made the selection?

Line 109: where there any restrictions or requirements as farm to be a candidate for the animal welfare assessment?

Response: The criteria for the inclusion of the evaluated farms was added to M&M, which included the voluntary participation of producers, because in Colombia, due to the armed conflict and public order problems, producers are reluctant to submit detailed information on their farms. There are also problems due to robbery and extortion, an aspect that we did not mention in the article, but which limited participation in the project. In addition, sampling costs are high and funding resources are scarce (lines 135-142).

Line 129: ‘The animals were kept in groups without social mixing during the entire fattening process’: it is not clear when the fattening period starts and ends. Please add this information. Thus, before the fattening period the animals were mixed a few times or once?

Response: The information was added and clarified (lines 161-166)

Line 133: I don’t think the title is correct. Is it?

Response: the information was corrected (line 169).

Line 134: ‘A trained veterinarian…’ Thus, 1 person visited all the farms? So, there can be an observer effect? How was the veterinarian trained? Was his or her training validated?

Response: the information was clarified (see line 170-172, 240-245). A second group visited other farms, but this information was not presented, because they were not interested in it. A panel selected the measures.

Line 162-164: ‘After completing the farm visits, a critical evaluation of…’: remove the word ‘critical’ because the word is superfluous, you must be always critical…

Response: It was removed (line 240).

Line 162-164: ‘…was carried out with the participation of two international experts…’: Why 2 experts? Can you justify the number of experts? Should be 1 expert enough? Or should a panel be better? And what is the definition of an expert? When can you say he or her is an expert (what characterises the experts (he is trained, years of experience, he is a researcher,…)?

Response: During September 2023, an evaluation panel was held to discuss the second version of the initial protocol, a process in which the authors of this article participated, and for this reason, we have included the results of this process in this article, in compliance with the requirements made by the evaluators (see lines 240-245). We added the second version of the methodology in the references.

Line 170: I miss some information, for e.g. about the weather conditions, the time of observation (the start and stop time)… These parameters can have an impact on the results and hence on your protocol.

Response: This information was added (see lines 174-179)

Line 171: ‘Statistical analysis’: how did you know that the sample size was sufficient for statistical analysis? Did you do a power analysis before the start of the assessment? If not, it is still interesting to do it.

Response: We did not validate the protocol, only test the feasibility of the proposed measures according to suggestions made by the reviewers. This apart was changed (lines 234-239).

Line 171: it is not clear why you used statistical analysis: to include or to exclude indicators? Clarify in the text.

Response: It was clarified (lines 234-239).

Line 176-182: this is not information about statistical analysis. It is necessary information, but not under the title ‘2.5 Statistical analysis’.

Response: It was changed as suggested (lines 234-260). Thank you for your help…

Line 189: ‘Table 3’: in the column on the right: at the first welfare criteria you write ‘Subjective assessment…’ but not at the last welfare criteria (lameness, complementary care, human-animal interaction…). Are these not assessed subjectively? Is there a difference between the assessment of the (first and the last) criteria? If it is, what is the difference?

Response: They are similar, the information was added (table 3)

Line 193: ‘Table 4’: adjustments of measures: ‘interview on…’: gives an interview objective information to include in the protocol? A respondent can say anything during an interview, but is not always correct… Isn’t it better to exclude these parameters than include information that is not validated for its correctness?

Response: You are absolutely in the right, moreover this information was easy to measure during the visits to the pastures, which is why this methodology was excluded from the table.

Line 196-199: ‘…the difficulty of measuring them on all farms, animal welfare implications not very applicable to the fattening production system under pasture conditions, time and space limitations, measures requiring specialized evaluations and adjustments to place the indicator in a category, where it was more feasible to evaluate it’: are these valuable arguments to exclude them? Or are these excuses to exclude them? And why not exclude more indicators on this basis? In my opinion is a detailed argumentation to include or exclude an indicator strongly needed. 

Response: You are absolutely in the right, moreover this information was clarified as suggested (lines 247-249, 251-253; 342-348).

Line 205: ‘Table 5’/discussion: I’m not agree with the argumentation of the removal of the indicators (column right). I don’t believe that important welfare criteria as hunger and thirst can be excluded from a valuable animal welfare assessment protocol. If there is enough grass on the pasture, you can suggest the animals are not hungry, like you wrote. If there is a natural well in the neighbourhood (distance between animals and well) you could also suggest that the animals are not thirsty, etc.

Response: It was changed as suggested, this indicators were added (see tables 3 and 4)

Line 219: ‘Discussion’: The discussion is about the justification for excluding certain animal welfare criteria, but should rather be about solutions to include certain criteria anyway. How can you talk about valuable and reliable animal welfare assessment protocol when a lot of parameters are excluded.

Response: It was reviewed as suggested (see tables 3 and 4).

Reviewer 2 Report

Comments and Suggestions for Authors

Table 1

Pasture area or Farm size would be more appropriate term instead of extension

Table 2

Table needs to be formatted better to delineate which welfare criteria the animal welfare measures are aligned with. For example there is no separation between “Subjective assessment of shade in the paddocks” and “General condition of the paddocks” even though one is associated with thermal comfort and the other with Comfort around resting

Good Nutrition or Adequate Nutrition instead of good feeding

Supplementation based on nutritional analysis instead of bromatological analysis

Feed storage instead of Food storage

For Comfort around resting there is not a method of assessment listed

In general manuscript needs to be edited for translation to English. Many words used do not make intuitive sense with standard English terminology.

               Comfort around resting, scrapers, draught protection, watering and shade feeders, Animal rest

Line 164, criteria: (a) herd with 1000 animals… where is b? assume you had categories for herd size, they should be listed here or referenced in a table.

Line 166, need to define difference between calm and excitable, what was determination based on? If one animal was excited but the rest calm, some percentage.

Line 167, describe what the interaction with cattle was, did they ride through them, group them, herd them, separate them? Was it the same at each farm?

Line 169, define positive, negative and neutral and describe how that is applied to a herd rather than individual

Table 3, provide BCS scale, most 1-5 BCS scales I am familiar with would not consider <= 4 a thin animal, 4 is thin on US 1-9 beef cattle scale

Table 3, lameness category, posture would be more appropriate rather than aplombs

Line 223-233, are these are the analysis that was done, it may be useful to have a table

Line 310-315, is repeat of previous sentences

Live 461, is there supposed to a reference in the ()

Comments on the Quality of English Language

Overall, English is acceptable quality but there are several instances where the term commonly used in agriculture is not translated properly

Author Response

Table 1

Pasture area or Farm size would be more appropriate term instead of extension

Response: It was changed as suggested (see table 1).

Table 2

Table needs to be formatted better to delineate which welfare criteria the animal welfare measures are aligned with. For example there is no separation between “Subjective assessment of shade in the paddocks” and “General condition of the paddocks” even though one is associated with thermal comfort and the other with Comfort around resting

Response: It was changed as suggested (see table 2).

Good Nutrition or Adequate Nutrition instead of good feeding

Supplementation based on nutritional analysis instead of bromatological analysis

Feed storage instead of Food storage

For Comfort around resting there is not a method of assessment listed

Response: It was modified as suggested (see tables).

In general manuscript needs to be edited for translation to English. Many words used do not make intuitive sense with standard English terminology.

               Comfort around resting, scrapers, draught protection, watering and shade feeders, Animal rest

 Response: It was edited as suggested (see the new version of article).

Line 164, criteria: (a) herd with 1000 animals… where is b? assume you had categories for herd size, they should be listed here or referenced in a table.

Response: It was added to clarify the information (lines 135-142)

Line 166, need to define difference between calm and excitable, what was determination based on? If one animal was excited but the rest calm, some percentage.

Response: It was clarified (see lines 196-214)

Line 167, describe what the interaction with cattle was, did they ride through them, group them, herd them, separate them? Was it the same at each farm?

Response: It was added as suggested (lines 196-214)

Line 169, define positive, negative and neutral and describe how that is applied to a herd rather than individual

Response: It was clarified (lines 208-214)

Table 3, provide BCS scale, most 1-5 BCS scales I am familiar with would not consider <= 4 a thin animal, 4 is thin on US 1-9 beef cattle scale

Response: It was corrected as suggested (see table 3).

Table 3, lameness category, posture would be more appropriate rather than aplombs

Response: It was changed as suggested (see table 3).

Line 223-233, are these are the analysis that was done, it may be useful to have a table

Response: The information was removed because it does not correspond to the objectives of the study. We added the descriptive analysis of results on farm (263-292).

Line 310-315, is repeat of previous sentences

Response: It was removed as suggested.

Live 461, is there supposed to a reference in the ()

Response: It was added (line 551)

Overall, English is acceptable quality but there are several instances where the term commonly used in agriculture is not translated properly

Response: It was reviewed as suggested (see all article). Thank you for your help.

Reviewer 3 Report

Comments and Suggestions for Authors

Dear authors,

I was ask to review you manuscript entitled "Validation of an animal welfare assessment protocol for Zebu beef farms within pasture-based systems under tropical conditions". The work includes an important topic and adresses the conflict, that pasture based system are not good by priniciple.

However, I have some concerns about this work. Althouhg you adress the background of validation in beginning of your discussion, you do not valdidate your protocoll. If any, you test the feasibilty of the measures of the protocoll. But you do not test if this protocoll at all is able to test the welbeing of the animals, for this you need any kind of comparison ("gold standard").

Another major concern is, I miss your results. Why did you not publish the findings of your farm visitis. You shoud have data on this. Variability of these data would be of interest. Furthermore I did not understand the use of your correlation testing.

In the discussion you repeat more or less your finding of the results. It becomes clear, that your protocoll is especially tested for the respective region you were doing your investigations.  Please consider to discuss this in a large context. As a remark, "not comming close to the animals" should not be a critieria to exclude measures, as this is a measure itself (human animal relationship).

I made further comments in the pdf, and hope they are helpful for you.

Author Response

However, I have some concerns about this work. Althouhg you adress the background of validation in beginning of your discussion, you do not valdidate your protocoll. If any, you test the feasibilty of the measures of the protocoll. But you do not test if this protocoll at all is able to test the welbeing of the animals, for this you need any kind of comparison ("gold standard").

Response: Your comment was very important for us. We changed the title, the objective, M&M and results (title, 12-13; 24-26; 88-98; 234-304, 295-300…. )

Another major concern is, I miss your results. Why did you not publish the findings of your farm visitis. You shoud have data on this. Variability of these data would be of interest. Furthermore I did not understand the use of your correlation testing.

Response: The results of the farm visits were added and the descriptive analysis was made (lines 263-292)

In the discussion you repeat more or less your finding of the results. It becomes clear, that your protocol is especially tested for the respective region you were doing your investigations.  Please consider to discuss this in a large context. As a remark, "not comming close to the animals" should not be a critieria to exclude measures, as this is a measure itself (human animal relationship).

Response: The discussion and your comment were reviewed (See tables 3 and 4, discussion)

I made further comments in the pdf, and hope they are helpful for you.

PDF COMMENTS

Line 46:  essenstial is maybe a strong word here

Response: It was changed as suggested (line 53).

Line 66: maybe a clear definition of pasture based system in discrimination to confined systems would be helpful. Are feedlots confined? what is the minimum pasture for an animal required to be pasture based?

Response: this information was added, see lines 78-81.

Line 81: Could you please specifiy the overall aim of this welfare assessment. Is it a regulatory tool for minimun standards of animal welfare or is it a to to categorize animal products from differnt production conditions with more or less animal welfare?

Response: this information was added (see lines 88-98).

Line 131: Did they received any concentrates or minerals? or other feed aditives

Response: this information was added (see lines 165-166).

Line 164: is more than a) coming?

Response: This information was added (lines 192-195)

Line 182-183: what was the idea of testing the association between 2 measures? so if age of employee would correlate to lameness on the farm (just by chance) what would be the consequence?

Response: The information was removed because it does not correspond to the objectives of the study.

Line 198: In 2.5 statistical analysis you mentioned 4 categories..now there are only three.

Response: It was corrected as suggested (lines 255-260).

line 201-202 (table.3): classification is missing in these indicators , below you presented classifactions. 

Response: It was added as suggested (see table 3)

this can be a very transient criteria. If you visit a farm just after the animals led onto a new pasture (maybe because an accessor is about to come)

Response: You are absolutely in the right, moreover a proportional random sampling was performed to controlled it.

complementary care 2. High: dot instead of slash

Response: It was changed as suggested (see table 3).

Line 204-205-206 (table4): euthanasia

Response: It was changed as suggested (see line 310).

Line 228-229-230-231-232-233-234: here it is a bit unclear, where you found the correlations, as spearman tests only correlation between 2 factors

Response: You are absolutely in the right. We only found correlation between 2 factors. This information was removed because it does not correspond to the objectives of the study according to suggestions.

As a validation of this protocol I expected some data on reproducability of validityy to measure welfare...but it seems you tested the feasablity of the various measures

Response: You are absolutely in the right. We applied your suggestion and changed the title, objective, M&M…(please see all document)

Line 249-250-251-252-253-254: I totally agree, please see comment at end of results

Response: It was changed as suggested.

Line 259: Maybe use another term. Protocoll seemed to be the complete assessment protocol

Response: It was changed as suggested, We replace it for methodology (see lines 15, 301, etc).

Line 280: please consider, that the protocol is not only for pasture based cattle in Magdalene Medio but in sub tropical conditions..as the titel suggests

Response: Thank you a lot…... It was changed as suggested (lines 372-373)

Line 294: again..should be included ...when assessment protocoll is used in other the the magdalene medio región

Response: It was changed as suggested (line 356).

Line 335: did you consider the evaluation of potentila resting areas (shade, dry) with category yes/no for the respective number of animals on the pasture (they will probably stay together )

Response: I forget that in the article, but the methodology included it. This information was added in the new version of the article. Thanks…..(see table 3)

Line 349: as a result of your data or literature ..then please add reference

Response: It was added as suggested: Bewley et al., 2017. A 100-year review: Lactating dairy cattle housing management  https://doi.org/10.3168/jds.2017-13251 (line)

Line 365: in some countries, farmer have to have an official training for the use of anaestheic...so this could be easy to measure

Response: You are right. It was added as suggested (see table 3).

Line 397-398-399: but you described the assessment of the cattle on horse back, which is in general a safe method to approach the animal. However, I believe it is important to assess the animals while they are on pasture in their envireoment. If you would put them in a crush, it would shadow lame animals and ohter painful situations

Response: It was changed as suggested (table 3)

Line 418-419: would you account snake bites and predator losses as welfare issues? I would encounter them as natural dangers in a natural envireoment

Response: It was removed as suggested. This information is not important for animal welfare.

Line 428-429-430-431-432-433: It is unclear, what is the message of this paragraph? As pasture based systems are the requirement for these protocols

Response: It was clarified.

Line 435: paddock or pasture?

Response: It was corrected as suggested (line 533).

Line 441-442: is this the condition for the feasablity of the measure. I think a good human animal relationsship is a valid indicator for animal welfare as well. Maybe you could rephrase this paragraph

Response: You are quite right and according to the criteria established to evaluate the feasibility of the measures, the evaluation of the animals in the pens would not meet a number of these criteria (additional handling of the animals and time and space constraints).

Reviewer 4 Report

Comments and Suggestions for Authors

Although I am a researcher in animal welfare, I do not have cattle-specific knowledge, so I cannot speak to the novelty of this study in the field, but it would seem that a validated welfare assessment of cattle for slaughter is of great importance to ensuring their quality of life.  However, I can speak to the quality of the validation process of the welfare assessment in this study.  Broadly I feel that this study is scientifically sound and thoughtfully written, though I do have a few primary concerns.  Namely, the qualifications or inclusionary criteria of the experts used in the study should be included in the manuscript.  This is important to ensure that they are indeed experts in the field.  My other primary concern is a seeming lack of attention given to the overall welfare and quality of life of this cattle population, as evidenced by only a brief mention of a lack of pain mitigation in routine health/medical practices.  This would seem to significantly affect cattle welfare, and thus is deserving of much greater attention in the manuscript.  Lastly, the formatting of the tables is not ideal for reader comprehensibility.  The within table text should be revised to ensure consistency in descriptions, and if possible, the tables should be horizontally oriented to allow for better display of the qualitative data.  Below are all of my recommendations for revision:

Methods

Line 78:  is a particular type of validation (e.g. construct validity) being sought in this study?

Lines 89-91:  it might be worthwhile to state in which language(s) farmers were informed about the study, as well as which language the informed consent document was written in.

Line 96:  only one expert was involved in the development of the evaluation protocol?  An “expert in animal welfare” is quite a broad term; it might be useful to add more detail if possible, such as the welfare of what species.

Lines 111-112:  how were these specific farms selected to participate in the study?  The recruitment process of farms should be described in more detail.  If these are all of the beef farms in the region, rather than being selected from a larger pool of farms, then that should be stated.

Line 113:  the range in herd size seems considerable and noteworthy; it would be useful to note the average herd size (or other measures of central tendency) of the participating farms.

Table 1:  typo – should be “altitude”.

Line 130:  since “at full milk” is in quotation marks, this suggests that it’s not a widely known term in the field; if so, it would be helpful to provide a brief definition of the term.

Line 138:  the subheading (Animal of Animal Welfare) seems to not make sense – perhaps there’s a typo?

Line 139:  is this the welfare protocol that was described in subsection 2.2?  If so, then it would be helpful to clarify this by using the protocol’s name.

Table 2:  the overall format of the table is poor and difficult to understand – perhaps due to typesetting issues.  It needs considerable reformatting to make it more comprehensible.

Table 2:  it might be helpful to state who was interviewed for some of the welfare measures/indicators.

Table 2:  similar to above, what type of calculation was conducted for some of the welfare measures/indicators.

Line 139:  did only one veterinarian conduct all of these assessments?  That seems like quite a lot of work for a single person in a relatively short period of time.

Line 150:  consider including the questionnaire in the appendix.

Lines 152-157:  are these all factors that were addressed in the questionnaire?  If so, it might be better to reword the beginning of the sentence to clarify that.  (e.g. “These factors, amongst others, were addressed in the questionnaire…”)

Line 172:  what type of experts were these or what qualifications did they have (e.g. bovine welfare researchers)?

Lines 189-193:  it might be helpful to make the four guidelines easier to follow if they were written in a list format below the paragraph, rather than within the paragraph.

Results

Table 4:  possible typo -  should “HAZARDS” be fully capitalized?

Tables 3, 4, and 5:  both are somewhat hard to follow due to formatting and inconsistencies in wording of text, such as in the ‘Method of Assessment’ column.  Because these tables contain qualitative data, and are thus rather large, it would be ideal if they could be horizontally (rather than vertically) oriented.

Line 229:  in which direction were these correlations?  

Discussion

Lines 249-256:  there doesn’t seem to be any mention of promoting overall good quality of life for cattle as a benefit of welfare assessments.  Even if this hasn’t been reported as a benefit in other relevant studies, based on the development and implementation of welfare assessments and standards with other animal species (e.g. domestic dogs, laboratory animals), it would seem worthwhile and important to note here.

Lines 295-302:  perhaps this is outside the scope of this study, but it would be useful to have assessments of body conditions and other factors assessed by observation to have been conducted by multiple experts, in order to establish the inter-rater reliability of this assessment means.

Lines 367-376:  in the interest of ensuring overall good welfare of these cattle, a much greater emphasis should be placed on the finding that common livestock practices are often performed without pain mitigation.  A lack of pain mitigation for routine or non-routine health or veterinary practices in other animal species would be considered inhumane and negatively impacting quality of life by relevant experts, so it would be logical that it should be equally applied to this cattle population.  Although this finding is consistent with other research, it nonetheless is noteworthy and deserving of greater attention in the Discussion.  Similarly, the suggestions for improvement in this area provided by the authors should be more urgent.

Other notes

It is surprising that within the ‘appropriate behavior’ principle, the ability to perform normal species-specific behaviors is not included in the welfare criteria.  Similarly, there was no mention of the cattle having agency/the ability to make their own choices in the welfare criteria.

Author Response

  • Namely, the qualifications or inclusionary criteria of the experts used in the study should be included in the manuscript.  This is important to ensure that they are indeed experts in the field.

Response: It was clarified as suggested (lines 170-173, 240-245)

  • My other primary concern is a seeming lack of attention given to the overall welfare and quality of life of this cattle population, as evidenced by only a brief mention of a lack of pain mitigation in routine health/medical practices.  This would seem to significantly affect cattle welfare, and thus is deserving of much greater attention in the manuscript.  

Response: It was complemented as suggested (lines 328-339)

  • Lastly, the formatting of the tables is not ideal for reader comprehensibility.  The within table text should be revised to ensure consistency in descriptions, and if possible, the tables should be horizontally oriented to allow for better display of the qualitative data.  Below are all of my recommendations for revision:

Response: Tables were modified as suggested (see tables)

  • Methods

Line 78:  is a particular type of validation (e.g. construct validity) being sought in this study?

Response: It was changed according to the suggestion of other evaluators. We test the feasibility of the animal welfare measures; we did not do a validation. We changed and complemented the article.

 Lines 89-91:  it might be worthwhile to state in which language(s) farmers were informed about the study, as well as which language the informed consent document was written in.

 Response: To clarify this point, we do not consider it relevant to state that it was in Spanish because the evaluation methodology was prepared in Spanish, which is the official language in Colombia. We did not translate any document from English for this process.

Line 96:  only one expert was involved in the development of the evaluation protocol?  An “expert in animal welfare” is quite a broad term; it might be useful to add more detail if possible, such as the welfare of what species.

Response: It was changed as suggested (see lines 294-300)

Lines 111-112:  how were these specific farms selected to participate in the study?  The recruitment process of farms should be described in more detail.  If these are all of the beef farms in the region, rather than being selected from a larger pool of farms, then that should be stated.

 Response: this information was clarified as suggested (lines 135-142)

Line 113:  the range in herd size seems considerable and noteworthy; it would be useful to note the average herd size (or other measures of central tendency) of the participating farms.

Response: It was complemented as suggested (lines 192-195)

Table 1:  typo – should be “altitude”.

Response: It was changed as suggested. Thank you (see table 1).

Line 130:  since “at full milk” is in quotation marks, this suggests that it’s not a widely known term in the field; if so, it would be helpful to provide a brief definition of the term.

Response: This information was clarified (lines 161-166)

Line 138:  the subheading (Animal of Animal Welfare) seems to not make sense – perhaps there’s a typo?

Response. This information was changed as suggested (line 169).

Line 139:  is this the welfare protocol that was described in subsection 2.2?  If so, then it would be helpful to clarify this by using the protocol’s name.

Response: It was added (lines 127-128).

2:  the overall format of the table is poor and difficult to understand – perhaps due to typesetting issues.  It needs considerable Table reformatting to make it more comprehensible.

 Response: it was changed as suggested (all tables).

Table 2:  it might be helpful to state who was interviewed for some of the welfare measures/indicators.

 Response: this information was added in M&M (line 220).

Table 2:  similar to above, what type of calculation was conducted for some of the welfare measures/indicators.

 Response: Table 3 describe it.

Line 139:  did only one veterinarian conduct all of these assessments?  That seems like quite a lot of work for a single person in a relatively short period of time.

Response: It was described as suggested (lines 170-172, 240-245).

 Line 150:  consider including the questionnaire in the appendix.

 Response: The questionnaire was written in Spanish and We have problems to traduce all document. Please excuse us.

Lines 152-157:  are these all factors that were addressed in the questionnaire?  If so, it might be better to reword the beginning of the sentence to clarify that.  (e.g. “These factors, amongst others, were addressed in the questionnaire…”)

 Response: It was changed as suggested (line 221- 222)

Line 172:  what type of experts were these or what qualifications did they have (e.g. bovine welfare researchers)?

Response: this information was added as suggested (line 117; 240-245).

Lines 189-193:  it might be helpful to make the four guidelines easier to follow if they were written in a list format below the paragraph, rather than within the paragraph.

Response: It was changed as suggested (line 255-259)

Results

 Table 4:  possible typo -  should “HAZARDS” be fully capitalized?

 Response: It was changed as suggested

Tables 3, 4, and 5:  both are somewhat hard to follow due to formatting and inconsistencies in wording of text, such as in the ‘Method of Assessment’ column.  Because these tables contain qualitative data, and are thus rather large, it would be ideal if they could be horizontally (rather than vertically) oriented.

Response: It was changed as suggested (tables).

 Line 229:  in which direction were these correlations?  

 Response: This information was removed for suggestion of other evaluator.

Discussion

Lines 249-256:  there doesn’t seem to be any mention of promoting overall good quality of life for cattle as a benefit of welfare assessments.  Even if this hasn’t been reported as a benefit in other relevant studies, based on the development and implementation of welfare assessments and standards with other animal species (e.g. domestic dogs, laboratory animals), it would seem worthwhile and important to note here.

 Response: It was added as suggested (lines 329-339). Thank you for your help.

Lines 295-302:  perhaps this is outside the scope of this study, but it would be useful to have assessments of body conditions and other factors assessed by observation to have been conducted by multiple experts, in order to establish the inter-rater reliability of this assessment means.

 Response: This suggestion is very important, and We will development this methodology in other studies (lines 557-579).

Lines 367-376:  in the interest of ensuring overall good welfare of these cattle, a much greater emphasis should be placed on the finding that common livestock practices are often performed without pain mitigation.  A lack of pain mitigation for routine or non-routine health or veterinary practices in other animal species would be considered inhumane and negatively impacting quality of life by relevant experts, so it would be logical that it should be equally applied to this cattle population.  Although this finding is consistent with other research, it nonetheless is noteworthy and deserving of greater attention in the Discussion.  Similarly, the suggestions for improvement in this area provided by the authors should be more urgent.

 Response: You are right. Excuse me. It was added as suggested (457-465; 584-589)

Other notes

It is surprising that within the ‘appropriate behavior’ principle, the ability to perform normal species-specific behaviors is not included in the welfare criteria.  Similarly, there was no mention of the cattle having agency/the ability to make their own choices in the welfare criteria.

Response: You are right, and as the protocol is in the process of being overhauled, your valuable comments will be considered. Thank you for helping us to look at cattle differently, perhaps, we have become accustomed to seeing them suffer.

Round 2

Reviewer 1 Report

Comments and Suggestions for Authors

Dear authors,

The text has been thoroughly reworked. Thank you for your efforts. I have no further comments.

Regards

Comments on the Quality of English Language

/

Reviewer 2 Report

Comments and Suggestions for Authors

Manuscript has been much improved from original

Line 17, animal herds or lots

Line 197, I don’t really like using the term batch, recommend group or lot, (here and elsewhere)

Line 281, delete they